# The Relationship between Korean Parents’ Smartphone Addiction and That of Their Children: The Mediating Effects of Children’s Depression and Social Withdrawal

**DOI:** 10.3390/ijerph19095593

**Published:** 2022-05-05

**Authors:** Shin-Il Lim, Sookyung Jeong

**Affiliations:** 1Department of Educational Psychology, College of Nursing, Jesus University, Jeonju 54989, Korea; imsi@jesus.ac.kr; 2Department of Nursing, College of Medicine, Wonkwang University, Iksan 54538, Korea

**Keywords:** smartphone addiction, parent–child relationship, parenting style, depression, social withdrawal

## Abstract

With the number of smartphone users growing around the world, children are using smartphones from an increasingly early age. Consequently, a significant number of children are being exposed to the risk of smartphone addiction, which is emerging as a serious social problem. Smartphone addiction can negatively impact children’s physical, cognitive, and social development. Previous studies have demonstrated that parental smartphone addiction influences that of their children. Therefore, this study explores the relationship between parental smartphone addiction and children’s smartphone addiction and the mediating effects of children’s depression and social withdrawal. Data are drawn from National Youth Policy Institute’s 2018 Korean Children and Youth Panel Survey. Respondents comprise 2011 fourth-grade elementary school students and their parents. Data were analyzed using SPSS version 21.0 and AMOS 21.0 software. Results show that the relationship between parental smartphone addiction and that of their children has a significantly positive mediating effect on children’s social withdrawal, but no such effect on children’s depression and there were no serial effects of children’s depression and social withdrawal. Consequently, educational programs that control parents’ smartphone usage, improve the parent–child relationship, and engender social sensitivity should be developed to reduce and prevent smartphone addiction among children.

## 1. Introduction

The global number of smartphone users is increasing, with smartphone usage steadily growing among children. Compared to other screening devices, including televisions, smartphones are frequently used for educational purposes among young children [1], such as reading books and watching educational programs. Consequently, the majority of children are being exposed to smartphones at an early age. Indeed, 38.7% of Argentine children between the ages of two and four can use mobile screens without help from their parents [2], while some 46% of American children are believed to have begun using smartphones below the age of two [3]. Research indicates that Korean children typically start using smartphones at 4.6 years old [4]. According to the Korean Statistical Information Service [5], 96.7% of South Korean children aged 10–19 use smartphones, with 27.3% classified as high-risk for smartphone addiction [6].

Kang and Park (2012) defined smartphone addiction as a disorder in which everyday life is affected due to nervous anxiety caused by excessive immersion in smartphones [7]. Lin et al. (2014) identified four main factors: compulsive behavior, tolerance, withdrawal, and functional impairment [8]. Previous studies proved that smartphone addiction in children is connected to their mental health, including depression [9], emotion and quality of life [10], self-esteem, parents’ smartphone addiction [11,12,13,14], and parental attitude [15]. Taking these into account, we can infer that it can negatively impact their life; thus, necessitating various interventions [16]. Several studies have shown that parents are strong influencing factors on children’s smartphone addiction [8,9,10,11]. Among these factors, parental smartphone addiction has the most profound impact on children. According to family systems theory, each family member can be a subsystem within the family and members influence each other [17]. Considering this, the influence of parental smartphone addiction on their children needs to be explored further.

Research has confirmed the mediating effect of children’s depression on the relationship between parental and children’s smartphone addiction. More specifically, scholars have observed a serial mediating effect: parental smartphone addiction influences parenting style [18] or parental rejection [19], which affects children’s depression and causes children’s smartphone addiction. Parents with a smartphone addiction spend their time mostly chatting or using social networking services. Thus, they cannot attentively care for their children [20]. Moreover, it was found that they showed negative parenting styles including rejection and neglect as compared to parents who did not have a smartphone addiction [21]. As negative parenting style is known to weaken the relationship between parents and children and challenges stable bonding between the parties, it leads to the development of depression in children [22]. Consequently, failing to establish a stable relationship with their parents, children immerse themselves in a virtual relationship through smartphones [23,24]. Based on existing literature, we can expect the mediating effect of depression among parental smartphone addiction and their children’s smartphone addiction.

Social sensitivity is defined as “attention, prominence, and emotions involved in processing information about social assessment and social status” [25]. A previous study suggests that it is an important phenomenon in peer interaction among children [26], as it represents the ability to admit mistakes, accept others as they are, make others feel good, and be a flexible thinker. Although research has yet to confirm whether parental smartphone addiction affects children’s social withdrawal, parental “technoference” has been shown to influence Chinese adolescents’ smartphone addiction, with social sensitivity having a mediating effect on the relationship between the variables [27]. The term “technoference” was coined in reference to the daily disruption of interpersonal interaction or time spent together caused by digital and mobile technology [28]. Similarly, people with smartphone addiction have characteristics such as they do not communicate with others or interact with their family and friends due to overuse of smartphones [7]. Technoference and smartphone addiction point to the problem of interruption in interpersonal interaction due to overuse of smartphones, although interference includes a broader spectrum of electronic devices including wearables, tablets, and other mobile devices. Accordingly, the term infers smartphone addiction. Several other studies have demonstrated that social withdrawal and smartphone addiction are closely linked [29,30].

Despite the rapid increase in children’s smartphone addiction, few studies have explored how parental smartphone addiction affects that of their children. Moreover, although there are individual studies that indicate that parental smartphone addiction affects depression among children and causes social withdrawal, which further leads to smartphone addiction in children, there are a few studies that systematically verify this relationship. In addressing these gaps, this study analyzes the effect of parental smartphone addiction on children’s smartphone addiction among fourth-grade elementary schoolchildren, and discerns the mediating effect of children’s social withdrawal and depression in this regard.

## 2. Research Questions

This study examines the mediating effects of children’s depression and social withdrawal in the relationship between parental and children’s smartphone addictions (Figure 1). More specifically, this study evaluates first the direct effect of parental smartphone addiction on children’s smartphone addiction. Second, it confirmed the mediating effects of children’s depression and their social withdrawal. Finally, we tested the serial mediating effects of parental smartphone addiction, children’s depression, children’s social withdrawal, and children’s smartphone addiction. In doing so, this study seeks to answer the following three research questions:Is the structural model suitable for parental smartphone addiction, children’s depression, social withdrawal, and smartphone addiction?What kind of direct effects are at play in the relationship between parental smartphone addiction, children’s social withdrawal, depression, and smartphone addiction?Do children’s depression and social withdrawal mediate the relationship between parental smartphone addiction and children’s smartphone addiction?

## 3. Methodology

### 3.1. Study Participants

This study used data collected through the first wave of the Korean Children and Youth Panel Survey 2018 (hereinafter, KCYPS 2018) [31], a longitudinal study conducted by the National Youth Policy Institute. For sampling, a multi-stage stratified cluster sampling was used to construct the original sample using four steps. In the first step, 17 cities and provinces in South Korea, 6040 schools nationwide, and 471,566 students were identified as the population. In the second step, stratified sampling was used, which can provide population representativeness with a small sample size. In the third step, sample allocation was proportionally distributed based on the number of students to 17 cities and provinces, assuming an average of about 70% effective responses. In the fourth and final step, the surveyed schools were selected using probability proportional to size sampling (PPS sampling).

To achieve the aims of this study, we selected only fourth-grade elementary school students for two reasons. First, although missing values occur over time due to the characteristics of the panel data, there are almost no missing values in the first year, thereby ensuring that it is representative of the characteristics of the population. Second, as the fourth grade of elementary school is the middle grade between the lower and upper grades, they represent the characteristics of elementary school students well.

More specifically, this study selected 2607 fourth-grade elementary school students (1313 males and 1294 females) from the panel survey. As this study examines the influences of the variables related to children and their parents’ smartphone addiction, the sample was narrowed to a total of 2011 students (968 males and 1054 females) who owned smartphones and whose parents or primary caregivers owned smartphones. Accordingly, this study comprises a total of 4022 participants, namely, 2011 students and 2011 parents.

### 3.2. Measurements

#### 3.2.1. Parental Smartphone Addiction

This study used the smartphone addiction proneness scale for adults developed by [32] Kim et al. This scale comprises 15 items: five items related to disturbance in daily life (e.g., “excessive use of smartphones reduces work ability”), two items related to seeking virtual worlds (e.g., “When I can’t use my smartphone, I feel like I’ve lost the whole world”), four items related to smartphone withdrawal symptoms (e.g., “Without a smartphone, I feel restless and anxious”), and four items related to tolerance regarding overusing smartphones (e.g., “It is a habit to spend a lot of time on a smartphone”). Items are measured on a four-point Likert scale, ranging from 1 (“not at all”) to 4 (“strongly agree”). The higher the score, the higher the addiction to smartphones. The reliability of the original scale as measured by Cronbach’s alpha was 0.814, while the scale of this study has a reliability of 0.801.

#### 3.2.2. Children’s Smartphone Addiction

This study uses the smartphone addiction proneness scale developed by Kim et al. [21] to evaluate children’s smartphone addiction. As in the parental smartphone addiction scale above, this scale comprises five subdomains: five items related to disturbance in daily life, two items related to seeking virtual worlds, four items related to smartphone withdrawal symptoms, and four items regarding tolerance about smartphone usage. However, the scales differ in respect to the items pertaining to disturbance in daily life, with the children’s scale containing items such as “School grades drop due to excessive use of smartphones.” The higher the score, the higher the addiction to smartphones. The reliability of Yoon’s study (2021) as measured by Cronbach’s alpha was 0.875 [18], and this scale has a reliability of 0.824.

#### 3.2.3. Children’s Depression

To examine children’s depression related to smartphone addiction, this study surveyed a total of 10 items by modifying and supplementing the existing items related to the KCYPS [31]. Items (e.g., “I have no energy”) were measured on a four-point Likert scale ranging from 1 (“not at all”) to 4 (“strongly agree”). Higher scores are indicative of depression. The reliability of original scale as measured by Cronbach’s alpha was 0.923, whereas the scale of this study scale has a reliability of 0.893.

#### 3.2.4. Children’s Social Withdrawal

This study used a modified and supplemented version of Kim and Kim’s [33] scale to evaluate children’s social withdrawal. The scale comprises a total of five items, including “It is awkward to have a lot of people around me.” Items are measured according to a four-point Likert scale, ranging from 1 (not at all) to 4 (strongly agree). A higher score means a higher degree of social withdrawal. The reliability of original scale as measured by Cronbach’s alpha was 0.940, while scale of this study scale has a reliability of 0.857.

### 3.3. Data Analysis

Data were analyzed using SPSS Version 21.0(IBM Corp., New York, NY, USA) and AMOS 21.0 software(IBM Corp., New York, NY, USA). Following basic statistical analysis and confirming the normal distribution of the data, Confirmatory Factor Analysis (CFA) was performed to verify the fit of the measurement model. In accordance with this study’s theoretical framework, a model was established based on the relationship between potential variables, and the fit was verified through chi-square (χ^2^), Comparative Fit Index (CFI), Tucker Lewis Index (TLI), and Root Mean Square Error of Approximation (RMSEA) analyses. As χ^2^ is sensitive to the number of samples, CFI and TLI were analyzed based on a value of 0.90 or higher and RMSEA was analyzed based on a value of 0.80 or less [34]. Finally, to examine the mediating effect on the relationship between elementary school student and parental smartphone addictions, children’s depression, children’s social withdrawal, and children’s smartphone addiction, bootstrap sampling was performed 2000 times using the Panton variable, confirming a significance level of 5% using a 95% confidence interval.

## 4. Results

### 4.1. Descriptive Results

Table 1 presents the mean, standard deviation, minimum, maximum, skew, and kurtosis of the variables analyzed in this study. Descriptive statistics of the variables indicate that the mean and standard deviation ranged from 1.547 to 2.048 (0.423 to 0.738). The minimum value of all variables was 1, and the maximum value was 3.76–4.0. Additionally, as for psychometric purposes, skewness and kurtosis values between −2 and +2 are acceptable [35,36]. The significance probability (P) of the Kolmogorov–Smirnov Z-value was higher than 0.05 [37]. Synthetically, the assumption of normality was satisfied.

### 4.2. Validation of the Fitness for the Measurement Model

The fitness index of the measurement model in this study was χ^2^ (DF) = 1476.5848 (224), *p*-value = 0.0000, CFI = 0.9348, TLI = 0.9263, and RMSEA = 0.0527; therefore, the model is suitable. The equality of each variable was confirmed by verifying convergent validity and discriminant validity. Convergent validity is verified if the standard estimate is 0.50 or more, Average Variance Extracted (AVE) is 0.5 or more, and composite reliability (CR) is 0.70 or more [38]. Table 2 presents the result of convergent validity. In respect to the standardized regression weights of each observed variable (β), parental smartphone addiction was 0.6079–0.7690, children’s depression was 0.5362–0.7871, children’s social withdrawal was 0.6269–0.8201, and children’s smartphone addiction was 0.7214–0.7693. All the variables were significant at *p* < 0.001.

Convergent validity was confirmed through AVE values, with all latent variables found to be over 0.50 and CR values over 0.70. As discriminant validity indicates the difference between each latent variable, it can be verified by comparing AVE and the square of the correlation between construct concepts [38]. Table 3 shows the correlations between construct concepts based on CFA.

### 4.3. Verification of Research Model

The structural equation model analysis was conducted to examine the structural causal relationships between parental smartphone addiction, children’s depression, children’s social withdrawal, and children’s smartphone addiction. Figure 2 presents the research model.

This study used the maximum likelihood method as the parameter estimation method. Table 3 presents the multicollinearity of parental smartphone addiction, children’s depression, children’s social withdrawal, and children’s smartphone addiction. As the tolerance was 0.7061–0.9750, which is close to 1, the VIF was 1.0257–1.4161, and all variables were below 10, multicollinearity was satisfied. The fitness of the research model was χ^2^ (DF) = 1476.5848 (224), *p*-value = 0.0000, CFI = 0.9348, TLI = 0.9263, and RMSEA = 0. Therefore, the model is suitable. As this model is an equivalent to the measurement model in the previous section, it shows the same goodness of fit.

Table 4 presents the pathways of the research model. Based on Figure 2 and Table 4, the significant pathways of the research model and each pathway coefficient are as follows. First, parental smartphone addiction had a significant positive effect on children’s depression (β = 0.0661, *p* < 0.05), children’s social withdrawal (β = 0.0681, *p* < 0.05), and children’s smartphone addiction (β = 0.1589, *p* < 0.001). Second, children’s depression did not have a significant impact on children’s social withdrawal and children’s smartphone addiction. Third, children’s social withdrawal had a significant positive effect on children’s smartphone addiction (β = 0.2882, *p* < 0.001).

Based on the results presented in Table 4, the mediating effect of children’s social withdrawal in the relationship of parental smartphone addiction and children’s smartphone addiction was examined using bootstrapping. As Table 5 shows, children’s social withdrawal had a significantly positive mediating effect (0.0204, *p* < 0.001) in the relationship between parental smartphone addiction and children’s smartphone addiction.

## 5. Discussion

In the growing IT industry, smartphones have become a must-have item and an essential good in everyday life [39]. Consequently, regardless of age, most people own smartphones, resulting in an increasingly large number of people with a smartphone addiction. Previous studies have identified and verified the risk factors associated with children’s smartphone addiction, including the length of time of smartphone use, parental styles, and parental smartphone addiction. According to Lin et al. [40] and Sözbilir and Dursun [41], the length of time children spend using smartphones is closely related to smartphone addiction. As smartphone usage increases, so does the likelihood of smartphone addiction. Such children can retain this addiction in adulthood, making it important to control and guide children’s smartphone usage.

Parents are believed to play a vital role in reducing children’s smartphone addiction. For example, children’s smartphone addiction has been found to decline with higher levels of parental risk perception about mobile devices and restrictive mediation [42]. Meanwhile, negative parenting style [43] and parental neglect [44] have been found to exacerbate children’s smartphone addiction. Additionally, several studies have demonstrated that parents’ smartphone addiction influences that of their children [45,46,47]. The results of these studies align with our findings. Smartphone addiction in adults has also been linked to a tendency to isolate and seldom engage in conversation with their family [48], as well as a lack of self-control [49]. Consequently, if parents are addicted to smartphones, they are able to control neither their own nor their children’s smartphone usage. Therefore, parents’ smartphone addiction should be considered as a risk factor of children’s smartphone addiction, suggesting the need for intervention programs that educate parents and children together.

In this study, the mediating effect of social withdrawal revealed the path through which parental smartphone addiction leads to children’s smartphone addiction (Figure 2). Examining the same context, Liu et al. [27] similarly found that parental technoference is related to their children’s smartphone addiction, with social sensitivity having a mediating effect. Social withdrawal refers to various responses including nervousness and withdrawal when encountering unfamiliar persons or environment, and difficulties in forming and maintaining good relationships with people, which lead to emotional and social maladaptation [50,51]. Previous studies have established the negative effects of social withdrawal in childhood. Significantly, social withdrawal lasts for a long time, leading to social anxiety, social phobias, and depression [51], as well as difficulties in peer relationships [52]. More specifically, adults with smartphone addiction tend to be less interested and neglectful in caring for their children, leading to issues in the parent–child relationship [53,54]. Consequently, given the importance of the parental role, children whose parents are addicted to their smartphones may be left alone more frequently and experience difficulties in social development. Furthermore, even though there is limited empirical evidence to suggest the optimal amount of time for children to be exposed to screens and efficacious parenting behaviors related to limiting children’s screen time, research indicates that positive parent–child relationships and effective family communication serve to reduce children’s problematic smartphone use [55] and social withdrawal [56]. In sum, the parent–child relationship plays a key role in preventing children’s social withdrawal and smartphone addiction. However, this study did not evaluate the parent–child relationship and parenting behavior as mediating effects. Therefore, further research must test how parenting behavior, parent–child relationship, and social withdrawal influence the relationship between parental smartphone addiction and their children’s smartphone addiction.

This study explored the structural relationship between parental smartphone addiction, children’s depression, and children’s smartphone addiction. Results show that children’s depression does not play a mediating role in the relationship between parental smartphone addiction and their children’s addiction. This finding contradicts those of other studies [18,57]. These studies clarified serial mediating effects of parenting style and children’s depression and the relationship between parental smartphone addiction and their children’s smartphone addiction. The higher the level of smartphone addiction, the higher the negative parenting behaviors such as overprotection, rejection, and neglect. Whereas, the lower the level of smartphone addiction, the higher the positive parenting behaviors such as encouragement and warmth [58]. Parents with a smartphone addiction tend to have a negative parenting style, since they do not have much time to care for their children due to their smartphone addiction [48,49]. Nonetheless, negative parenting behaviors worsen the parent–child relationship, which is linked to their children’s depression [59,60]. As the children were unable to have a stable relationship with their parent, they tend to immerse themselves in the virtual world to form relationships with others, which consequently leads to smartphone addiction [23]. However, there are some studies that suggest that smartphone addiction can be a predicting factor of depression as people with higher smartphone addiction showed higher depression scores [61]. Consequently, considering bidirectionality among two variables, further research needs to confirm the direction of the correlation for children.

This study has several limitations. First, this study used children’s self-reporting data, which means that there is a possibility that data do not correctly reflect respondents’ attitudes. Additionally, as respondents were fourth-grade elementary students, data are not representative of all Korean children. Therefore, further research covering various age cohorts is necessary. Second, the results of this study cannot be generalized to middle and high school students because they may be less influenced by their parents. Accordingly, a follow-up study is needed to determine how the results of this study differ from those pertaining to middle and high school students. Third, as this study utilized secondary data of a panel survey, the influence of variables not included in the analysis cannot be excluded. This needs to be taken into account when interpreting the results of this study. To address this limitation, further study including more variables that influence smartphone addiction in adults and children is necessary. Additionally, since this study used panel data, we cannot include hours of parental smartphone usage for examining parental smartphone addiction symptoms. Therefore, further research needs to include a quantitative description of parental smartphone usage to evaluate parents’ own perceptions about smartphone overuse.

## 6. Conclusions

This study demonstrates that parental smartphone addiction can influence children’s smartphone addiction, with children’s social withdrawal found to have a mediating effect on the relationship between parental and children’s smartphone addiction.

This study can contribute to understanding of the influence of parental smartphone addiction on their children. In this study, we found that it is important to control not only children’s smartphone usage but also that of their parents. Therefore, it is necessary to develop educational programs that control both parents’ and their children’s smartphone usage in school and community centers. Additionally, this study verified that parental smartphone addiction increased children’s social withdrawal, which led to their smartphone addiction. However, as previous studies proved that parental behaviors or parenting styles and children’s social withdrawal had serial mediating effects on the relationship between parents’ and children’s smartphone addiction, further studies must be conducted to better understand the systematic relationships among these variables. Additionally, individual/group counseling and education to improve the parent–child relationship and social sensitivity should be conducted and implicated. As children with smartphone addiction are more likely to develop depression within a few years, we recommend conducting a follow up cohort study on this group.

## Figures and Tables

**Figure 1 ijerph-19-05593-f001:**
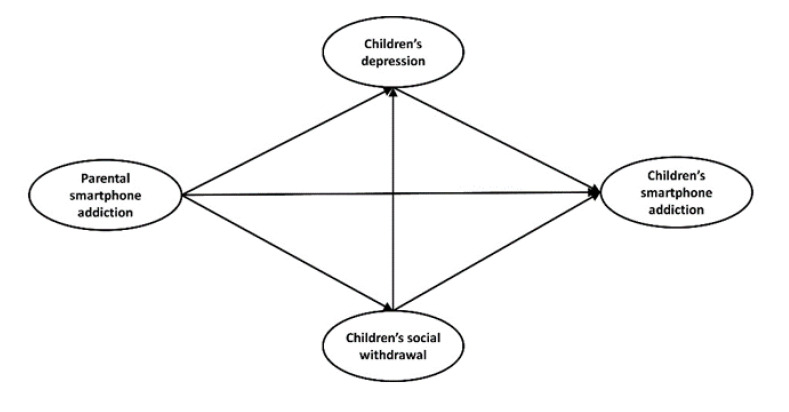
Hypothesized research model.

**Figure 2 ijerph-19-05593-f002:**
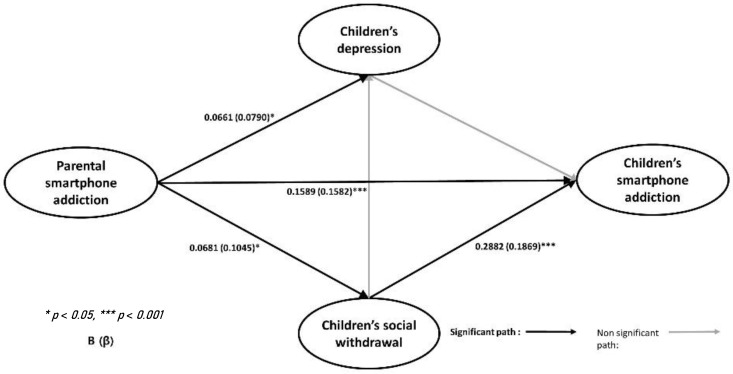
Path coefficients for the research model.

**Table 1 ijerph-19-05593-t001:** Statistics of variables (*n* = 2011).

	Mean	SD	Min	Max	Kolmogorov–Smirnov Z(*P*)	Skew	Kurtosis
Parental smartphone addiction	1.7365	0.4235	1.0000	3.7625	0.6404(0.776)	0.4639	0.2336
Children’s depression	1.5473	0.5383	1.0000	4.0000	0.9728(0.230)	1.1204	1.2136
Children’s social withdrawal	2.0489	0.7389	1.0000	4.0000	0.5119(0.878)	0.2825	−0.6415
Children’s smartphone addiction	1.7756	0.4955	1.0000	4.0000	0.6303(0.807)	0.6877	0.5308

**Table 2 ijerph-19-05593-t002:** Analysis of convergent validity.

	Unstandardized Coefficients (B)	Standardized Coefficients (β)	Standard Error	t	AVE	CR
PSA→tolerance	1.0000	0.7690 *			0.5086	0.8048
PSA→withdrawal	0.9091	0.7498 *	0.0312	29.2783		
PSA→virtual world	0.8264	0.7128 *	0.0293	28.2954		
PSA→disturbance	0.7249	0.6079 *	0.0296	24.5779		
CD→CD1	1.0000	0.6877 *			0.5711	0.8982
CD→CD2	1.1068	0.7871 *	0.0351	31.9411		
CD→CD3	1.0805	0.6244 *	0.0429	25.8182		
CD→CD4	0.8582	0.6804 *	0.0312	27.9794		
CD→CD5	0.8871	0.5362 *	0.0403	22.3583		
CD→CD6	1.0289	0.6488 *	0.0385	26.7732		
CD→CD7	0.9967	0.7174 *	0.0346	29.3795		
CD→CD8	0.8575	0.7043 *	0.0304	28.8818		
CD→CD9	0.8325	0.6664 *	0.0304	27.4462		
CD→CD10	1.0112	0.7742 *	0.0324	31.4737		
CSW→CSW1	1.0000	0.7073 *			0.5502	0.8587
CSW→CSW2	1.1109	0.8201 *	0.0345	32.8343		
CSW→CSW3	0.9864	0.7367 *	0.0333	30.0393		
CSW→CSW4	1.1352	0.7981 *	0.0357	32.1941		
CSW→CSW5	0.8497	0.6269 *	0.0332	25.8332		
CSA→disturbance	1.0000	0.7324 *			0.5467	0.8281
CSA→SVW	0.9477	0.7214 *	0.0333	28.7590		
CSA→withdrawal	0.9881	0.7693 *	0.0334	30.2454		
CSA→tolerance	1.1673	0.7322 *	0.0400	29.1237		

Notes: * *p* < 0.001. PSA = parental smartphone addiction; tolerance = tolerance about using smartphones; SVW = seeking virtual world; disturbance = disturbance in daily life; CD = children’s depression; CSW = children’s social withdrawal; CSA = children’s smartphone addiction.

**Table 3 ijerph-19-05593-t003:** Relationship between construct concepts, convergent validity, and multicollinearity.

		PSA	CD	CSW	CSA
PSA		1.0000			
CD		0.1053 *	1.0000		
CSW		0.0647 *	0.5138 *	1.0000	
CSA		0.1440 *	0.2977 *	0.2696 *	1.0000
convergent validity	AVE	0.5086	0.5711	0.5502	0.5467
	composite reliability	0.8048	0.8982	0.8587	0.8281
multicollinearity	tolerance	0.9750	0.7061	0.7211	0.8804
	VIF	1.0257	1.4161	1.3868	1.1358

Notes: * *p* < 0.001. PSA = parental smartphone addiction; CD = children’s depression; CSW = children’s social withdrawal; CSA = children’s smartphone addiction; VIF = variance inflation factor.

**Table 4 ijerph-19-05593-t004:** Pathways of the research model.

Pathway of Variables	Standardized Coefficients (β)	Unstandardized Coefficients (B)	Standard Error	CR
PSA→CD	0.0661 *	0.0790	0.0311	2.5385
PSA→CSW	0.0681 *	0.1045	0.0409	2.5551
PSA→CSA	0.1589 ***	0.1582	0.0265	5.9677
CD→CSW	0.0071	0.0091	0.0324	0.2829
CD→CSA	−0.0184	−0.0153	0.0207	−0.7436
CSW→CSA	0.2882 ***	0.1869	0.0174	10.7268

Notes: * *p* < 0.05, *** *p* < 0.001. PSA = parental smartphone addiction; CD = children’s depression; CSW = children’s social withdrawal; CSA = children’s smartphone addiction.

**Table 5 ijerph-19-05593-t005:** Verification of indirect effects.

Pathway of Variables	Indirect Effect	95% Confidence Interval
Parental smartphone addiction → children’s social withdrawal → children’s smartphone addiction	0.0204 *	0.0058–0.0364

Note: * *p* < 0.001.

## Data Availability

The following are available online at https://www.nypi.re.kr/archive/mps/program/examinDataCode/dataDwloadAgreeView?menuId=MENU00226 (accessed on 2 February 2022).

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
