# Peer review of "The Relationship between Korean Parents’ Smartphone Addiction and That of Their Children: The Mediating Effects of Children’s Depression and Social Withdrawal"

_ijerph, 2022, doi:10.3390/ijerph19095593_

Round 1

Reviewer 1 Report

In the abstract section, please report the sampling method.

In the introduction, please write previous research on the topic under study and explain the research gap. Please explain more about the research variables such as depression and social withdrawal and their relationship with smartphones and the impact of this study on the spread of science needs further explanations. Overall, the problem of the statement on the research issue was weak and should be reconsidered.

In the method part, the sampling method was ambiguous and it was not clear whether the whole community was selected or sampled. Please be clear. Also please report the validity and reliability of the tools in other research. 

In the result part, for normal distribution, kurtosis and skewness are weak indicators. Please report the results of the Kolmogorov Smirnov test.

The discussion part was repetitive explanations and the explanations were similar to the introduction. Interpretation of results should be based on the results of previous research and theories presented in the field under study.

Author Response

The manuscript has been rechecked and the necessary changes have been made in accordance with the reviewers’ suggestions. The responses to all comments have been prepared and attached here with/given below. The revisions have been written in red in the text.

Thank you for your consideration. I look forward to hearing from you

Reviewer 2 Report

The paper aims to identify the mediating effect of children’s social withdrawal and depression in the relationship between parents’ and children’s smartphone addiction. The topic is significant though not pathbreaking. The discussion offers valuable insights for readers.

Though the paper mentions several studies in the relevant fields, please include a short literature review to help set the context of the study and identify gaps.  A mention of previous studies that have used the same scales as in this study will also provide better context to the findings. For example, if a study used the smartphone addiction proneness scale for adults, what did their findings show and how do they reflect on the present findings?

The theoretical concept or framework is not discussed at length probably because structural equation models have become quite commonplace. The researchers may consider including a brief summary of the model to benefit readers. There is a need to describe the research process so that the readers are able to appreciate how the responses were recorded. The questionnaire scales are appropriate and suited to the research design. 

The data analysis procedure, measures taken to establish reliability and suitability of the SEM model are described in detail. The main results show that parental smartphone addiction had a significant positive effect on children’s depression, social withdrawal and smartphone addiction. Children’s social withdrawal had a significant positive effect on their smartphone addiction. These are important findings and the researchers have showcased this relationship. However, their true significance should be further highlighted by linking them to further studies that can show the ills of childhood depression and smartphone addiction.

 The findings have practical merit for practitioners in child psychology as well as policy makers. However, as the researchers have not highlighted the way forward in terms of further studies or contribution of the study at length, this should be addressed.

The paper needs further proofreading and editing to improve the flow of the paragraphs and present the information better. The addition of more related studies, contribution of the study, and implications for researchers and practitioners will add to the readability of the paper.

Author Response

The manuscript has been rechecked and the necessary changes have been made in accordance with the reviewers’ suggestions. The responses to all comments have been prepared and attached herewith/given below. The revisions have been written in red in the text.

Thank you for your consideration. I look forward to hearing from you

Please check the attachment

Reviewer 3 Report

This study of the relationships between parents´and their children´s smartphone addiction as well as between addiction on one hand and depression and withdrawal in the  children on the other hand is based upon an ongoing panel study in Korea and comprises 2011 parents and their 2011 fourth grade children in elementary school.

In general it is well written although there are occasional language errors which need attention. The study is cross-sectional and this has not been sufficiently discussed by the authors. In such a study it is always difficult to know the direction of the correlations. It seems odd to assume that smartphone addiction in the child could cause smartphone addition in the parent, but such a causality counter-direction might be relevant in the discussion. For instance, bidirectional thinking may be of value when we discuss the lack of association between child smartphone addiction and child depression. A child who develops a depression due to such addiction might be the subject of substantial efforts from parents and others resulting in decreased depression despite continued smartphone addiction. Such counter- processes might reduce the correlation down to null between addiction and depression.

The authors also rightly point out that teachers, class mates and other persons could influence the situation, but that is another matter.

Perhaps this cohort will be followed for a couple of years. It would really be interesting to see prospective results. It could be for instance that smartphone-addiction related depression in these growing people could arise after a delay of two years. 

Statistically the authors have the necessary knowledge although I was a bit confused by the fact that they seemed to accept a high threshold (below 3) for labelling "normal distribution" although the highest skewness they report in their study variables is 1.21 which is ok.

The use of a large number of decimals always raises suspicions in me. More than two decimals are seldom warranted and more than three is not ok for me. Four and five decimals give me the impression that the authors want to look more scientific than they are.

Author Response

Dear Reviewer

The manuscript has been rechecked and the necessary changes have been made in accordance with the reviewers’ suggestions. The responses to all comments have been prepared and attached herewith/given below. The revisions have been written in red in the text.

Thank you for your consideration. I look forward to hearing from you

Reviewer 4 Report

Thank you for submitting this important and interesting research article. Overall, the paper highlights the important influence of parent smartphone use, and its links to children's smartphone use, and psychological wellbeing.

Please see attached a document with my comments and suggestions. Overall, the paper would benefit from providing a clearer discussion of the key constructs, an acknowledgement of the emerging understandings of the issues discussed, and directions for research in future. 

I congratulate the authors on their important work. I hope the comments provided are beneficial in revising your manuscript.

Author Response

Dear reviewer

The manuscript has been rechecked and the necessary changes have been made in accordance with the reviewers’ suggestions. The responses to all comments have been prepared and attached herewith/given below. The revisions have been written in red in the text.

Thank you for your consideration. I look forward to hearing from you

Round 2

Reviewer 3 Report

The authors have adequately addressed my concerns, in particular the point about the cross-sectional nature of the study which precludes advanced conclusions about causality.